# Non-Vacuous Generalization Bounds: Can Rescaling Invariances Help?

## Abstract

A central challenge in understanding generalization is to obtain non-vacuous guarantees that go beyond worst-case complexity over data or weight space. Among existing approaches, PAC-Bayes bounds stand out as they can provide tight, data-dependent guarantees even for large networks. However, in ReLU networks, rescaling invariances mean that different weight distributions can represent the same function while leading to arbitrarily different PAC-Bayes complexities. We propose to study PAC-Bayes bounds in an invariant, lifted representation that resolves this discrepancy. This paper explores both the guarantees provided by this approach (invariance, tighter bounds via data processing) and the algorithmic aspects of KL-based rescaling-invariant PAC-Bayes bounds.

## 1 Introduction

Deep neural networks generalize well despite being massively overparameterized, a fact that remains only partially explained by statistical learning theory (Zhang et al., 2021; Belkin et al., 2019; Bartlett et al., 2021). Among existing approaches, PAC-Bayes bounds are especially promising: they are *data dependent* and have yielded non-vacuous guarantees for large models (Dziugaite & Roy, 2017; Dziugaite et al., 2021; Pérez-Ortiz et al., 2021; Letarte et al., 2019; Biggs & Guedj, 2021; 2022a;b). A persistent limitation, however, is that standard PAC-Bayes analyses are carried out in *weight space* $\mathcal{W}$: the prior $P$ and posterior $Q$ are distributions on parameters $w \in \mathcal{W}$, and the complexity is typically a divergence such as the Kullback–Leibler (KL) one $D_{\mathrm{KL}}(Q\|P)$. For ReLU networks, neuron-wise rescaling symmetries imply that many parameterizations implement the same predictor $f_w$ while producing wildly different divergences. As a result, weight-space PAC-Bayes bounds can vary arbitrarily across functionally equivalent models.

**A motivating example.** Consider the one-hidden-neuron ReLU network $f_w(x) = w_2 \max(w_1 x, 0)$ with $w = (w_1, w_2) \in \mathbb{R}^2$. For any $\lambda > 0$, the rescaled parameters $\diamond^\lambda(w) := (\lambda w_1, w_2/\lambda)$ satisfy $f_{\diamond^\lambda(w)} = f_w$. If $P \sim \mathcal{N}(0, \sigma^2 I_2)$ and $Q \sim \mathcal{N}(w, \mathrm{diag}(w^2))$, then the rescaled posterior $\diamond^\lambda_\sharp Q$ induces a KL divergence $D_{\mathrm{KL}}(\diamond^\lambda_\sharp Q \| P) \sim \lambda^2 w_1^2 / \sigma^2$ when $\lambda$ tends to infinity, which can be made arbitrarily large although the predictor is unchanged. This simple case already shows that weight-space bounds are not aligned with functional equivalence.

**Two complementary routes toward invariance.** We adopt a viewpoint that makes rescaling invariance explicit and leads to a concrete program built around three questions.

*Route A: deterministic (and stochastic) rescaling in weight space.* A first natural idea is to keep working in $\mathcal{W}$ but to take the best bound over rescalings of the prior and posterior. Deterministic rescaling uses the group action $w \mapsto \diamond^\lambda(w)$ at hidden units; we later broaden this to *stochastic rescaling* that randomly rescales hidden units in a way that preserves $f$ almost surely.

*Route B: lifted (invariant) representations.* A second idea is to *lift* parameters to an intermediate space $\mathcal{Z}$ collapsing rescaling symmetries. Formally, consider a *rescaling-invariant* measurable map (a "lift") $\psi : \mathcal{W} \to \mathcal{Z}$ and a measurable $g : \mathcal{Z} \to \mathcal{F}$ such that $f_w = g(\psi(w))$. An instance of $\psi$ for ReLU networks[1] is the path+sign lift $\psi(w) = (\Phi(w), \mathrm{sign}(w))$, obtained by augmenting with the signs the so-called "path-lifting" $\Phi$, a path-based representation of the weights that appears, e.g., in

---

[1] Theorem 4.1 in Gonon et al. (2025) shows that $\psi(w) = \psi(w')$ implies $f_w = f_{w'}$. Hence $\psi$ is indeed a lift: defining $g : \mathrm{Im}(\psi) \to \mathcal{F}$ by $g(z) := f_w$ for any $w$ with $\psi(w) = z$ yields the factorization $f_w = (g \circ \psi)(w)$.

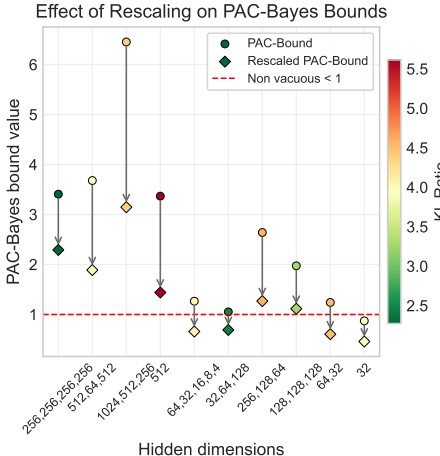 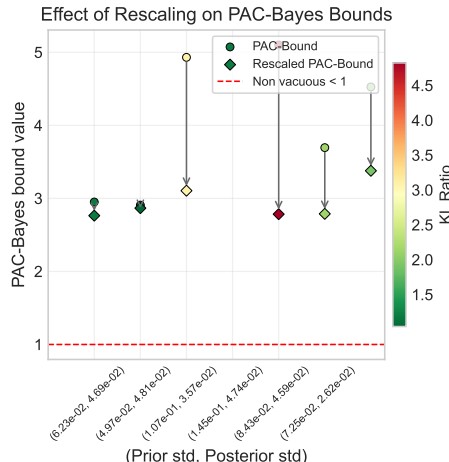

(a) PAC-Bayes bounds for **MLPs on MNIST**. Each vertical line = one architecture (hidden-layer widths on $x$-axis). Test accuracy: min $95.81\%$, mean $97.49\%$, max $98.13\%$.

(b) PAC-Bayes bounds for **CNN on CIFAR-10** ($86\%$ test accuracy). Each vertical line = one $(\text{prior std}, \text{posterior std})$ pair.

Figure 1: Impact of deterministic rescaling on PAC-Bayes bounds. **Left (MNIST)**: MLPs with varying hidden-layer widths. **Right (CIFAR-10)**: CNN with varying $(\sigma_{\text{prior}}, \sigma_{\text{posterior}})$. Circles: original bounds; diamonds: bounds optimized over deterministic rescaling (which is an upper bound on the lifted $D_{\text{KL}}$ by Equation (1)). The red dashed line marks the non-vacuous threshold ($< 1$).

Neyshabur et al. (2015); Kawaguchi et al. (2017); Barron & Klusowski (2019); Stock & Gribonval (2023); Bona-Pellissier et al. (2022); Gonon et al. (2024); Gonon (2024); Gonon et al. (2025). We then attempt to prove PAC-Bayes bounds with divergences between pushed-forward distributions, e.g., $D_{\text{KL}}(\psi_\sharp Q \| \psi_\sharp P)$.

These two routes give rise to the following three questions that structure the paper.

**Q1 — Validity (Section 3).** *Can we state standard PAC-Bayes bounds in a lifted space?* We show that it is indeed the case for KL-based PAC-Bayes bounds, the change-of-measure step (Donsker–Varadhan) applies *verbatim* to the pushed-forward pair $(\psi_\sharp Q, \psi_\sharp P)$ as soon as $\psi$ is measurable and $\psi_\sharp Q \ll \psi_\sharp P$ (which holds whenever $Q \ll P$). The same argument extends to $f$-divergences. For Wasserstein distances, we show that it suffices to assume that the factorizer $g$ is Lipschitz (so that Lipschitz losses remain Lipschitz in the lifted representation, i.e., after composition with $g$) (see Appendix B.2).

**Q2 — Comparison of bounds (Section 4).** *How do the lifted and rescaling-optimized bounds relate to the non-lifted one?* For any measurable, rescaling-invariant lift $\psi$, the data processing inequality yields
$$D_{\text{KL}}(\psi_\sharp Q \| \psi_\sharp P) \leq D_{\text{KL}}(Q \| P).$$

Introducing stochastic rescaling $\diamond^{\boldsymbol{\lambda}}$ (rescaling operator by a random $\boldsymbol{\lambda}$ while preserving $f$) and the deterministic special case $\diamond^\lambda$ (with a deterministic rescaling vector $\lambda$), we establish the chain

$$D_{\text{KL}}(\psi_\sharp Q \| \psi_\sharp P) \leq \inf_{\boldsymbol{\lambda}, \boldsymbol{\lambda}'} D_{\text{KL}}(\diamond_\sharp^{\boldsymbol{\lambda}} Q \| \diamond_\sharp^{\boldsymbol{\lambda}'} P) \leq \inf_{\lambda, \lambda'} D_{\text{KL}}(\diamond_\sharp^{\lambda} Q \| \diamond_\sharp^{\lambda'} P) \leq D_{\text{KL}}(Q \| P), \quad (1)$$

which compares, in one stroke, the *lifted*, *stochastic-rescaling*, *deterministic-rescaling*, and *non-lifted* KL terms. Thus, lifted bounds are never worse and can be strictly tighter when symmetries are effectively collapsed.

**Q3 — Computation (Section 5).** *What is tractable in practice?* In general, neither the lifted KL nor the stochastic-rescaling infimum admits a closed form, even for Gaussian $(P, Q)$. By contrast, the *deterministic* infimum $\inf_{\lambda, \lambda'} D_{\text{KL}}(\diamond_\sharp^\lambda Q \| \diamond_\sharp^{\lambda'} P)$ is a computable upper-bound proxy for the two harder terms in Equation (1). We devise an algorithm with *global convergence* to this infimum,

via a hidden strict convexity that appears after an appropriate reparameterization. Empirically, this optimization yields smaller KL terms (e.g., typically $\sim \times 4$ smaller in Figure 1) and, consequently, tighter PAC-Bayes bounds (e.g., typically $\sim \times 2$ smaller in Figure 1, turning some vacuous bounds into non-vacuous ones).

**Outline.** Section 2 recalls the setting and notation (PAC-Bayes theory, rescaling invariances for ReLU networks). Section 3 establishes the lifted PAC-Bayes bounds (validity). Section 4 introduces stochastic rescaling[2] and proves the comparison chain (1). Section 5 develops the algorithm for the deterministic rescaling infimum and discusses the intractability of the lifted and stochastic-rescaling terms, along with experiments. Section 6 concludes and sketches directions for invariant, tractable priors directly in lifted space.

## 2 BACKGROUND

This section fixes notation and recalls the ingredients used throughout: (i) classical PAC-Bayes bounds (with a focus on KL in the main text), (ii) DAG–ReLU networks and their neuron-wise rescaling symmetry.

### 2.1 PAC-BAYES BOUNDS

PAC-Bayes theory (developed by Shawe-Taylor & Williamson, 1997; McAllester, 1998; 1999; Seeger, 2002; Catoni, 2007 – we refer to Guedj, 2019; Alquier, 2024; Hellström et al., 2025 for comprehensive introductions) provides data-dependent generalization guarantees for randomized predictors. Let $\ell : \mathcal{Y} \times \mathcal{Y} \to \mathbb{R}_{\geq 0}$ be a bounded loss, and let $f : \mathcal{W} \to \mathcal{F}$ map parameters $w \in \mathcal{W}$ to predictors $f_w \in \mathcal{F}$. For $w \in \mathcal{W}$, define the population and empirical risks

$$L(w) \; := \; \mathbb{E}_{(x,y)\sim\mathcal{D}}\big[\ell(f_w(x),y)\big], \qquad \hat{L}_S(w) \; := \; \tfrac{1}{n}\sum_{i=1}^{n} \ell\big(f_w(x_i),y_i\big), \tag{2}$$

associated with a distribution $\mathcal{D}$ on $\mathcal{X} \times \mathcal{Y}$ and a collection $S = \big((x_i,y_i)\big)_{i=1}^{n}$ of $n$ samples. The classical McAllester-type bound states that for any prior $P$ on the weights (fixed before observing the samples $S$), bounded loss $\ell \in [0,C]$ (e.g. $C = 1$ for the 0-1 loss in multi-class classification), $t > 0$ and $\delta \in (0,1)$, with probability at least $1 - \delta$ over $S \sim \mathcal{D}^{\otimes n}$, the following holds uniformly over all posterior $Q \ll P$ (so it might be chosen depending on $S$):

$$\mathbb{E}_{w\sim Q}\big[L(w)\big] \; \leq \; \mathbb{E}_{w\sim Q}\big[\hat{L}_S(w)\big] \; + \; \frac{t^2 C}{8n} \; + \; \frac{D_{\mathrm{KL}}(Q\|P) + \log(1/\delta)}{t} \tag{3}$$

which means that the generalization gap $L - \hat{L}_S$ averaged over the weight-posterior $Q$ can be controlled with the KL-divergence $D_{\mathrm{KL}}(Q\|P)$. Much of the literature tightens constants, relaxes assumptions, or replaces $D_{\mathrm{KL}}$ by other divergences ($f$-divergences, Wasserstein), see *e.g.* (Maurer, 2004; Catoni, 2007; Alquier & Guedj, 2018; Mhammedi et al., 2019; 2020; Biggs & Guedj, 2023; Picard-Weibel & Guedj, 2022; Clerico & Guedj, 2023; Haddouche & Guedj, 2023; Viallard et al., 2023; Adams et al., 2024; Hellström & Guedj, 2024; Clerico et al., 2025; Haddouche et al., 2025). In the main text we focus on KL-based, as doing so already exposes the issues and benefits of invariance and lifting; extensions are discussed in the appendix.

### 2.2 DAG–ReLU NETWORKS AND NEURON-WISE RESCALING

We consider the classical formalism of DAG–ReLU networks specified by a directed acyclic graph $G = (V, E)$ with input, hidden, and output neurons denoted respectively by $V_{\mathrm{in}}, H$ and $V_{\mathrm{out}}$ (Neyshabur et al., 2015; Kawaguchi et al., 2017; DeVore et al., 2021; Bona-Pellissier et al., 2022; Stock & Gribonval, 2023; Gonon et al., 2024). Parameters $w \in \mathcal{W} = \mathbb{R}^{E \cup (V \setminus V_{\mathrm{in}})}$ collect edge weights $w_{u \to v}$ and (optional) biases $b_v = w_v$ for $v \notin V_{\mathrm{in}}$. With ReLU activations, the network realization $f_w : \mathbb{R}^{|V_{\mathrm{in}}|} \to \mathbb{R}^{|V_{\mathrm{out}}|}$ is defined recursively by

$$v(w,x) = \begin{cases} x_v, & v \in V_{\mathrm{in}}, \\ \mathrm{ReLU}\Big(b_v + \sum_{u:\, u \to v} u(w,x)\, w_{u \to v}\Big), & v \notin V_{\mathrm{in}}, \end{cases} \qquad f_w(x) = (v(w,x))_{v \in V_{\mathrm{out}}}. \tag{4}$$

---

[2] and precisely defines the notation $\diamond_\sharp^\lambda Q$, which mimics the notion of pushforward

For simplicity, we omit pooling and identity neurons (which are often used to encode skip connections). Our results, however, extend directly to networks that include them; see Definition 2.2 in Gonon et al. (2024) for the formal class of DAG–ReLU networks covered.

**Deterministic rescaling.** Positive homogeneity of ReLU induces a neuron-wise rescaling symmetry. Let $H \subseteq V$ denote hidden neurons and let $\lambda = (\lambda_v)_{v \in H} \in \mathbb{R}_{>0}^{H}$, extended by $\lambda_v \equiv 1$ on $V \setminus H$. Define the (deterministic) rescaling operator

$$\diamond^{\lambda}(w) \text{ by } \left(\diamond^{\lambda}(w)\right)_{u \to v} = \frac{\lambda_v}{\lambda_u} w_{u \to v}, \qquad \left(\diamond^{\lambda}(w)\right)_{v} = \lambda_v w_v \tag{5}$$

where the operations are applied on the weights $w_e$ of the edges $e = u \to v$ as well as the biases $w_v = b_v$ of neurons. We will use $\diamond^{\lambda}_{\sharp} Q$ to denote the pushforward of a distribution $Q$ by $\diamond^{\lambda}$. Importantly we have $f_{\diamond^{\lambda}(w)} = f_w$ for every $w$.

**Stochastic rescaling.** We will also later consider *stochastic* rescaling $\diamond^{\boldsymbol{\lambda}}$ where $\boldsymbol{\lambda}$ is a random positive vector (Definition 2).

## 3 VALIDITY: PAC-BAYES BOUNDS IN LIFTED SPACES

PAC-Bayes bounds provide generalization guarantees for randomized predictors. Conceptually, the quantity of interest only depends on the *functions* realized by the network: one would ideally like to measure the discrepancy between the induced distributions of predictors, through a divergence $D(f_{\sharp}Q \,\|\, f_{\sharp}P)$ between the pushforwards of the posterior and prior in function space. Unfortunately, this ideal form is intractable in practice.

The standard workaround is to write PAC-Bayes bounds in terms of divergences between distributions over the *weights* themselves, $D(Q\|P)$, because these are often tractable (e.g., closed form for Gaussian priors/posteriors with KL). Yet this ignores symmetries: two parameter vectors $w, w'$ that realize the same function $f_w = f_{w'}$ are still treated as distinct in $D(Q\|P)$.

**Lifting the representation.** To address this, we consider measurable lifts $\psi : \mathcal{W} \to \mathcal{Z}$ satisfying the factorization property

$$f_w = g(\psi(w)) \qquad \text{for some measurable } g : \mathcal{Z} \to \mathcal{F}. \tag{6}$$

The lift may be chosen rescaling-invariant, but *invariance is not needed for validity*. Lifts can collapse weight-space redundancies and induce a funnel as in Figure 2

$$\mathcal{W} \xrightarrow{\psi} \mathcal{Z} \xrightarrow{g} \mathcal{F},$$

suggesting that divergences may shrink as one moves closer to function space.

**Can standard PAC-Bayes bounds, such as McAllester's classical result** (3)**, be established in terms of lifted divergences** $D(\psi_{\sharp}Q \,\|\, \psi_{\sharp}P)$**?**

**Answer: *yes*, by lifting the change of measure.** Our first contribution is to revisit the classical McAllester's bound and show that it can be stated directly in terms of any measurable lift. The key point is that the change-of-measure inequality underpinning PAC-Bayes proofs (the Donsker–Varadhan formula for KL) remains valid after lifting. Since the inequality only requires measurability of the loss and absolute continuity $\psi_{\sharp}Q \ll \psi_{\sharp}P$ (which holds whenever $Q \ll P$), the entire classical proof transfers verbatim (see Appendix A for details). We obtain the next lifted analogue of McAllester's bound:

**Proposition 1** (McAllester's bound in lifted space)**.** *Let $\psi : \mathcal{W} \to \mathcal{Z}$ be a measurable lift satisfying* (6)*. Let $P$ be a prior over weights, fixed before observing the samples $S$. For any $\delta \in (0,1)$ and $t > 0$, with probability at least $1 - \delta$ over $n$ i.i.d. samples $S$, the following holds uniformly over all $Q \ll P$:*

$$\mathbb{E}_{w \sim Q}[L(w)] \ \leq \ \mathbb{E}_{w \sim Q}[\hat{L}_S(w)] \ + \ \frac{t^2 C}{8n} \ + \ \frac{D_{\mathrm{KL}}(\psi_{\sharp}Q \,\|\, \psi_{\sharp}P) + \log(1/\delta)}{t}. \tag{7}$$

**Scope.** While we focus on McAllester's bound here since it is among the simplest PAC-Bayes results, the same underlying argument (lifted change-of-measure) extends to other KL-based bounds.

We focus on the KL-based bound above because it already highlights the benefits and obstacles of lifting. We also show that the same "lift-then-change-of-measure" template extends to other divergences used in PAC-Bayes in Appendix B:

- For $f$-divergences the corresponding variational forms carry over to $(\psi_\sharp Q, \psi_\sharp P)$ exactly as for KL (Appendix B.1 for details).

- For Wasserstein distances, one additionally requires that the generalization gap be Lipschitz in the lifted coordinates (e.g., via a Lipschitz assumption on $g$, see Appendix B.2).

In short, lifted PAC-Bayes bounds are established through lifted change-of-measures. This restores a form of representation-awareness when the lift absorbs invariance, while keeping the standard proof template intact. In the next sections we (i) compare lifted, stochastically/deterministically rescaled, and non-lifted KL terms, and (ii) develop a tractable proxy based on deterministic rescalings.

## 4 COMPARISON: LIFTED, RESCALED, AND NON-LIFTED KL

This section compares four KL terms that can appear in PAC-Bayes bounds: *(i)* the *lifted* KL $D_{\mathrm{KL}}(\psi_\sharp Q \| \psi_\sharp P)$ from Proposition 1, *(ii)* a *stochastically rescaled* (non-lifted) KL, *(iii)* a *deterministically rescaled* (non-lifted) KL, and *(iv)* the initial *non-lifted* KL. We show that these form a chain of inequalities, with the lifted term never larger than the others, and we clarify when (and how) one may optimize over rescalings without affecting the loss-dependent side of the bound.

### 4.1 DETERMINISTIC AND STOCHASTIC RESCALING

Recall the neuron-wise rescaling operator $\diamond^\lambda$ from Equation (5): for $\lambda \in \mathbb{R}_{>0}^H$ (extended by 1 on non-hidden units),

$$(\diamond^\lambda(w))_{u \to v} = \frac{\lambda_v}{\lambda_u} w_{u \to v}, \qquad (\diamond^\lambda(w))_v = \lambda_v w_v,$$

which preserves the realized function: $f_{\diamond^\lambda(w)} = f_w$.

**Deterministic rescaling of a distribution.** For a distribution $Q$ on $\mathcal{W}$, its deterministically rescaled version is $\diamond_\sharp^\lambda Q$, the pushforward of $Q$ by $\diamond^\lambda$.

**Stochastic rescaling (random, weight-dependent factors).** While *deterministic rescaling* preserves the induced function distribution, they are only a very special case of a more general family of *random* rescaling. For PAC-Bayes analysis, it is indeed natural to allow rescaling factors *themselves* to be random, and even to depend on the weights. This motivates the more general notion of *stochastic rescaling*.

**Definition 2.** Consider a random variable[3] $\boldsymbol{\lambda}$ potentially *dependent* on the random weights $w \sim Q$ (resp. $w \sim P$): in other words, $(\boldsymbol{\lambda}, w) \sim C$ with $C$ some joint distribution (or *coupling*). Given any draw $(\boldsymbol{\lambda}, w)$ the rescaled weights are defined as $w' := \diamond^{\boldsymbol{\lambda}}(w)$. This yields a *stochastic rescaling* of $w$, with distribution $w' \sim Q'$ and by a slight abuse of the *pushforward* notation we denote $\diamond_\sharp^{\boldsymbol{\lambda}} Q := Q'$ (resp. $w' \sim P' =: \diamond_\sharp^{\boldsymbol{\lambda}} P$).

For a fixed $\lambda$, if $(\boldsymbol{\lambda}, w) \sim \delta_\lambda \otimes Q$ then we recover the deterministic rescaling $Q' = \diamond_\sharp^{\boldsymbol{\lambda}} Q = \diamond_\sharp^\lambda Q$.

The next lemma shows that stochastic rescaling also preserves the induced distributions of functions, paving the way to further optimization of the KL term of McAllester's bound. It is the cornerstone to establish a sequence of bounds interpolating between the lifted bound of Proposition 1 and the non-lifted one of Equation (3).

**Lemma 3** (Function and lift invariance under stochastic rescaling). *Let $\psi$ be any rescaling-invariant lift (i.e., $\psi \circ \diamond^\lambda = \psi$ for all $\lambda$). For any distribution $Q$ on $\mathcal{W}$ and any (possibly weight-dependent) stochastic rescaling $\boldsymbol{\lambda}$,*

$$f_\sharp Q = f_\sharp(\diamond_\sharp^{\boldsymbol{\lambda}} Q), \qquad \psi_\sharp Q = \psi_\sharp(\diamond_\sharp^{\boldsymbol{\lambda}} Q). \tag{8}$$

---

[3]We use bold as a mnemonic to distinguish from deterministic rescaling $\lambda$

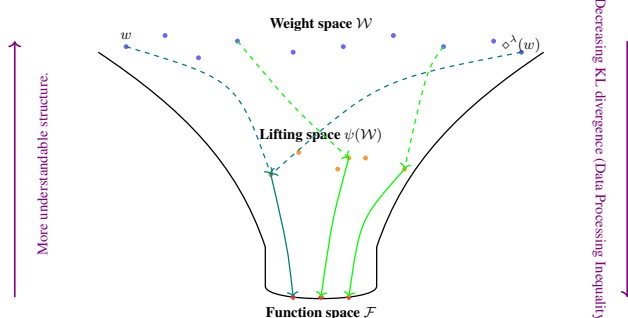

Figure 2: The information funnel $\mathcal{W} \to \mathcal{Z} \to \mathcal{F}$. Weight-space symmetries (e.g., rescaling) could be collapsed by the lift $\psi$, and the induced map to function space $f$ further compresses information. Divergences (e.g., KL) are expected to decrease along this chain, motivating the use of lifted-space bounds.

## 4.2 A CHAIN OF KL TERMS

Let $P, Q$ be prior/posterior distributions on $\mathcal{W}$, and let $\psi$ satisfy the factorization $f = g \circ \psi$ from Equation (6). By Lemma 3, $\psi_\sharp(\diamond_\sharp^{\boldsymbol{\lambda}} Q) = \psi_\sharp Q$ and $\psi_\sharp(\diamond_\sharp^{\boldsymbol{\lambda}'} P) = \psi_\sharp P$ for any stochastic rescalings $\boldsymbol{\lambda}, \boldsymbol{\lambda}'$. Applying data processing to the measurable map $\psi$ gives

$$D_{\mathrm{KL}}(\psi_\sharp Q \,\|\, \psi_\sharp P) \;=\; D_{\mathrm{KL}}\big(\psi_\sharp(\diamond_\sharp^{\boldsymbol{\lambda}} Q) \,\|\, \psi_\sharp(\diamond_\sharp^{\boldsymbol{\lambda}'} P)\big) \;\le\; D_{\mathrm{KL}}\big(\diamond_\sharp^{\boldsymbol{\lambda}} Q \,\|\, \diamond_\sharp^{\boldsymbol{\lambda}'} P\big).$$

Taking the infimum over stochastic rescalings and then restricting to deterministic ones yields the *comparison chain* (1) as follows:

$$D_{\mathrm{KL}}(\psi_\sharp Q \,\|\, \psi_\sharp P) \;\le\; \inf_{\boldsymbol{\lambda}, \boldsymbol{\lambda}'} \; D_{\mathrm{KL}}\big(\diamond_\sharp^{\boldsymbol{\lambda}} Q \,\|\, \diamond_\sharp^{\boldsymbol{\lambda}'} P\big) \tag{9}$$

$$\le\; \inf_{\lambda, \lambda'} \; D_{\mathrm{KL}}\big(\diamond_\sharp^{\lambda} Q \,\|\, \diamond_\sharp^{\lambda'} P\big) \;\le\; D_{\mathrm{KL}}(Q\|P).$$

The last inequality takes $\lambda = \lambda' = \mathbf{1}$.

In particular, the lifted divergence is never larger via data processing, and might actually be strictly smaller when symmetries are collapsed (it can even turn vacuous bounds to non-vacuous ones as we will observe in Figure 1). This formalizes the funnel intuition $\mathcal{W} \to \mathcal{Z} \to \mathcal{F}$ illustrated in Figure 2.

**Consequences for the PAC-Bayes bounds.** Combining the lifted bound Equation (7) with this chain of inequality shows that the same PAC-Bayes bounds but with $D_{\mathrm{KL}}(Q\|P)$ replaced by any of the three terms in Equation (9) yields a valid PAC-Bayes bound which is never larger than the original one. We study in the next section what can be computed.

## 5 COMPUTATION: WHAT IS (NOT) TRACTABLE, AND A PRACTICAL PROXY

The comparison chain (1) established in Section 4 (see Equation (9) above), suggests two natural computational routes beyond the raw weight-space KL: (i) push $P, Q$ through a rescaling-invariant lift $\psi$ and compute the *lifted* KL; (ii) optimize the *non-lifted* KL over rescalings (stochastic or deterministic). We now explain why the first two targets are challenging, and then develop a tractable and effective instance of the third one.

### 5.1 WHY THE LIFTED KL (WITH PATH + SIGN) IS CHALLENGING IN GENERAL

So far our discussion applied to any measurable lift $\psi$ (sometimes additionally assumed invariant). To make the lifted KL concrete, one must pick a specific lift. A lift that stands out in the literature is the *path + sign* lift $(\Phi(w), \mathrm{sign}(w))$, where $\Phi$ is the "path-lifting" which maps each weight vector to the collection of path products in the network.[4] This construction has played a central role in

---

[4]Strictly speaking, even though $\Phi$ is called path-lifting in the literature, it is not a lift in the sense of Equation (6); the sign component is needed to make it a lift, see Figure 6 in Gonon et al. (2025).

recent advances on identifiability (Stock & Gribonval, 2023; Bona-Pellissier et al., 2022), training dynamics (Marcotte et al., 2023), Lipschitz and norm-based bounds (Gonon et al., 2024; 2025), pruning (Gonon et al., 2025), and Rademacher-based generalization guarantees (Neyshabur et al., 2015; Barron & Klusowski, 2019; Gonon et al., 2024).

We observe that for this lift, even when $P, Q$ are simple (e.g., factorized Gaussians on edges/biases), computing $D_{\mathrm{KL}}(\psi_\sharp Q \,\|\, \psi_\sharp P)$ is challenging for two independent reasons:

**(i) Products already break closed forms.** A single coordinate of $\Phi(w)$ is a *product* of edge weights along a path (Gonon et al., 2024, Definition A.3). If edge weights are independent Gaussians, that product has a non-Gaussian law (computable only in the two-variable case, with a Bessel-type density) for which KLs rarely admit closed forms. Thus, even a *univariate* lifted KL term seems already out of reach.

**(ii) Path coordinates are dependent.** Two different paths can share edges. Their associated coordinates in the products $\Phi(w)$ therefore share terms, making the coordinates of $\Phi(w)$ *dependent* even if the coordinates of $w$ are independent. Therefore, the pushforwards $\psi_\sharp P$ and $\psi_\sharp Q$ do not factorize, and multivariate KLs cannot be reduced to sums of independent one-dimensional terms.

Together, (i) and (ii) make exact lifted KLs impractical beyond toy cases, even before accounting for the discrete sign part.

### 5.2 Why the stochastic-rescaling infimum is challenging

The middle term in the chain (9) optimizes over *stochastic* rescalings: $\boldsymbol{\lambda}$ may be random *and* depend on $w$. Even if $Q$ is Gaussian, the pushforward $\diamond_\sharp^{\boldsymbol{\lambda}} Q$ is then a *data-dependent random mixture of rescalings*, which has no simple parametric form in general; computing $\inf_{\boldsymbol{\lambda}, \boldsymbol{\lambda}'} D_{\mathrm{KL}}(\diamond_\sharp^{\boldsymbol{\lambda}} Q \,\|\, \diamond_\sharp^{\boldsymbol{\lambda}'} P)$ is therefore out of reach analytically, and challenging even numerically as it would require to optimize over the space of couplings $(\boldsymbol{\lambda}, w)$. Interesting questions left to future work include understanding whether the infimum is attained, how it could be approximated, and whether it coincides with the left-hand side $D_{\mathrm{KL}}(\psi_\sharp Q \,\|\, \psi_\sharp P)$.

### 5.3 Deterministic rescaling as a tractable proxy

Fortunately, the chain (9) includes a computable middle ground: the *deterministic* rescaling infimum

$$\inf_{\lambda, \lambda'} D_{\mathrm{KL}}\big(\diamond_\sharp^\lambda Q \,\|\, \diamond_\sharp^{\lambda'} P\big).$$

It upper-bounds the lifted KL and never exceeds the original weight-space KL. We now show it reduces to a one-sided problem and can be solved globally (for standard Gaussian priors), yielding a practical drop-in replacement in McAllester-style bounds.

**Theorem 4** (Optimized deterministic rescaling for zero-mean Gaussian priors)**.** *Let $G = (V, E)$ be a ReLU DAG with hidden neurons $H \subset V$, and let $\diamond^\lambda$ be the neuron-wise rescaling from Equation* (5)*.*

1. *(Reduction) For general $P, Q$ and any divergence $D(\cdot\|\cdot)$ satisfying the data processing inequality, the two-sided rescaling problem reduces to a one-sided one:*

$$\inf_{\lambda, \lambda' \in \mathbb{R}_{>0}^{|H|}} D\big(\diamond_\sharp^\lambda Q \,\|\, \diamond_\sharp^{\lambda'} P\big) = \inf_{\lambda \in \mathbb{R}_{>0}^{|H|}} J(\lambda) = \inf_{\lambda \in \mathbb{R}_{>0}^{|H|}} \bar{J}(\lambda). \tag{$\star$}$$

*where*

$$J(\lambda) := D\big(Q \,\|\, \diamond_\sharp^\lambda P\big) \quad and \quad \bar{J}(\lambda) := D\big(\diamond_\sharp^\lambda Q \,\|\, P\big), \quad \lambda \in \mathbb{R}_{>0}^{|H|} \tag{10}$$

2. *(Existence & uniqueness) If $D(\cdot\|\cdot) = D_{\mathrm{KL}}(\cdot\|\cdot)$, $P \sim \mathcal{N}(0, \sigma'^2 \mathbf{I})$, and $Q$ has finite second moments and admits a density with respect to the Lebesgue measure, then*

   (a) *$J$ admits a unique global minimizer $\lambda^\star$.*

   (b) *(Convergence of block coordinate descent) Consider the block coordinate descent (BCD) scheme that, given an order $(v_1, \ldots, v_{|H|})$ of the hidden neurons, cyclically updates one coordinate $\lambda_{v_\ell}$ at a time to its exact one-dimensional minimizer (which admits an analytical expression, see Algorithm 1 for a simple case, and Equation* (12) *for the general case). From any initialization $\lambda^{(0)} \in \mathbb{R}_{>0}^{|H|}$ the sequence $(\lambda^{(r)})_{r \geq 0}$ converges to $\lambda^\star$.*

*Consequently,*

$$D_{\mathrm{KL}}(\psi_\sharp Q \,\|\, \psi_\sharp P) \;\le\; \inf_{\boldsymbol{\lambda},\boldsymbol{\lambda}'} \; D_{\mathrm{KL}}\!\left(\diamond_\sharp^{\boldsymbol{\lambda}} Q \,\Big\|\, \diamond_\sharp^{\boldsymbol{\lambda}'} P\right) \;\le\; \underbrace{\inf_{\lambda\in\mathbb{R}_{>0}^{|H|}} J(\lambda)}_{\text{computable by BCD}} \;\le\; D_{\mathrm{KL}}(Q\|P),$$

*i.e., the deterministic-rescaling infimum is a* tractable upper bound *on the lifted-space KL and a* tighter proxy *than the original weight-space KL.*

The proof is given in Appendix C. The existence of a unique global minimizer for $P = \mathcal{N}(0, \sigma'^2 \mathbf{I})$ is due to the strict convexity of $z \in \mathbb{R}^{|H|} \mapsto J(\exp(z))$. The assumption on $P$ is not a strong constraint since it is very usual for a PAC-Bayes prior. The result remains valid for centered Gaussian $P$ with arbitrary diagonal covariance.

**Takeaway.** Exact lifted KLs (with path + sign) and stochastic-rescaling infima are generally intractable. The deterministic-rescaling infimum is a principled, tractable proxy: it upper-bounds the lifted KL, is never worse than the raw weight-space KL, and can be optimized globally (for common Gaussian priors) with a simple, fast BCD scheme.

## 5.4 ALGORITHM IN THE SIMPLE CASE, AND THE GENERAL NEURONWISE UPDATE

We first give the updates in a simple setup and refer to the appendix for the general formula. The proof in Appendix C.3 shows that convergence guarantees still apply if one updates in parallel any set of neurons such that no two of them are neighbors (otherwise their updates would interact). In layered fully-connected networks (LFCN), this allows *odd–even* parallel updates: rescale all odd layers simultaneously, then proceed similarly with even layers, and iterate until convergence.

**Square LFCN ($d$-by-$d$ matrices).** Let the network have depth $L$ and all layers (input, hidden, output) of width $d$. Denote by $\lambda_\ell \in \mathbb{R}_{>0}^d$ the rescaling vector of layer $\ell$. For a centered Gaussian prior $P \sim \mathcal{N}(0, \sigma'^2 \mathbf{I})$ and posterior $Q \sim \mathcal{N}(0, \sigma^2 \mathbf{I})$, all coordinates of $\lambda_\ell$ will have the same optimal coordinatewise update

$$\lambda_{\ell,k} \;\leftarrow\; \left(\frac{C_\ell}{A_\ell}\right)^{1/4}, \qquad k = 1, \dots, d, \tag{11}$$

where

$$A_\ell \;=\; \sigma^2 \sum_{j=1}^d \frac{1}{\lambda_{\ell+1,j}^2}, \qquad C_\ell \;=\; \sigma^2 \sum_{i=1}^d \lambda_{\ell-1,i}^2.$$

---

**Algorithm 1** Odd-even minimization of the KL over deterministic rescalings on a square LFCN for $P \sim \mathcal{N}(0, \sigma'^2 \mathbf{I})$ and $Q \sim \mathcal{N}(0, \sigma^2 \mathbf{I})$

---

**Require:** Stds $\sigma, \sigma' > 0$, sweeps $T$.
1: Initialize (implicitly) $\lambda_\ell \equiv \mathbf{1}_d$ for $\ell = 1, \dots, L$.
2: **for** $t = 1, \dots, T$ **do**
3:     **(Odd layers, in parallel)** For each odd $\ell \in \{1, 3, \dots\}$:
4:         Update $\lambda_{\ell,k} \leftarrow (C_\ell/A_\ell)^{1/4}$ for all $k = 1, \dots, d$         (by (11))
5:     **(Even layers, in parallel)** Same steps for even $\ell \in \{2, 4, \dots\}$.
6: **end for**
7: **Output:** Optimal $\lambda^\star$.

---

**General neuronwise update.** In general, Theorem 4 guarantees that the minimizer $\lambda^\star$ is reached by block coordinate descent. The generic algorithm updates the rescaling factor $\lambda_v$ of each neuron $v$ one by one (see Appendix C.3) as

$$\lambda_v \;\leftarrow\; \sqrt{\frac{-B_v + \sqrt{B_v^2 + 4A_v C_v}}{2A_v}}, \tag{12}$$

with $A_v, B_v, C_v$ given in Equations (14) to (16), which covers much more general $Q$, in particular non-centered and with distinct variances over distinct coordinates, as is often the case in traditional PAC-Bayes bounds. We deliberately keep these definitions in the appendix to avoid heavy notation here.

**Experiments.** We test our proxy on MNIST MLPs (input 784, output 10) with varying hidden-layer widths, ranging from $25K$ to $1.5M$ parameters for $55K$ training images, and on a CIFAR-10 CNN with about 5.2M parameters for $50K$ training images. For each model, we compare the standard PAC-Bayes bound (using $D_{\mathrm{KL}}(Q\|P)$) with its deterministic-rescaling version based on $\inf_{\lambda,\lambda'} D_{\mathrm{KL}}((\diamond^\lambda)_\sharp Q \,\|\, (\diamond^{\lambda'})_\sharp P)$. Figure 1 shows that rescaling typically halves bound values, turning some vacuous bounds into non-vacuous ones. These results confirm that deterministic rescaling can yield tighter and more practical guarantees. More details on the setups are given in Appendix D.

## 6 CONCLUSION

We studied PAC-Bayes generalization through the lens of rescaling invariances in ReLU networks. Lifting collapses symmetries and, by data processing, yields divergences that are never larger than in weight space. Our main practical contribution is a deterministic-rescaling proxy: it bounds from above the lifted KL, is never worse than $D_{\mathrm{KL}}(Q\|P)$, and can be computed via a globally convergent algorithm under standard Gaussian priors. Empirically, optimizing this proxy substantially tightens PAC-Bayes bounds, often turning vacuous guarantees into non-vacuous ones.

Via a chain of inequalities, we also showed the potential of tighter bounds associated to exact lifted KLs (e.g., path + sign) and stochastic-rescaling infima. Such bounds raise interesting mathematical and computational challenges, and are expected to catalyze new developments around invariant priors/posteriors and optimization schemes to bridge the remaining computability gap.

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

## A    MᴄAʟʟᴇꜱᴛᴇʀ's Bᴏᴜɴᴅ ɪɴ ᴛʜᴇ Lɪꜰᴛᴇᴅ Sᴘᴀᴄᴇ

The derivation of PAC-Bayes bounds in the *lifted space* hinges on a *change of measure* argument, especially leveraging the Donsker-Varadhan variational formula for KL-based PAC-Bayes bounds.

**Sketch.**    The lifted PAC-Bayes framework extends classical PAC-Bayes bounds by working in a *lifted space* $\mathcal{Z}$, obtained via the measurable map $\psi : \mathcal{W} \to \mathcal{Z}$. The proof proceeds in three main steps:

1. **Change of measure:** Apply the Donsker-Varadhan variational formula in $\mathcal{Z}$, exploiting the fact that pushforward distributions preserve absolute continuity.

2. **Pullback to the weight space:** Rewrite expectations and divergences in $\mathcal{Z}$ as expectations and divergences in $\mathcal{W}$ using the lift map $\psi$.

3. **Specialization to generalization error:** Instantiate the variational formula with the generalization error, and control the prior term via concentration inequalities.

The key insight is that the lifted structure allows us to derive bounds in $\mathcal{Z}$ while performing all computations in $\mathcal{W}$, preserving the interpretability and tractability of classical PAC-Bayes analysis.

**Details.**    *Step 1: Donsker–Varadhan variational formula in the lifted space.* Let $(\mathcal{W}, \mathcal{B})$ be a measurable space, and let $\psi : \mathcal{W} \to \mathcal{Z}$ be a surjective measurable map. The lifted space $(\mathcal{Z}, \sigma(\mathcal{B}, \psi))$ is a measurable space, where $\sigma(\mathcal{B}, \psi)$ is the final $\sigma$-algebra generated by $\psi$. For any probability distribution $P_{\mathcal{Z}}$ over $\mathcal{Z}$ and any measurable function $h : \mathcal{Z} \to \mathbb{R}$, the Donsker–Varadhan variational formula states:

$$\sup_{Q_{\mathcal{Z}} \ll P_{\mathcal{Z}}} \left( \mathbb{E}_{Z \sim Q_{\mathcal{Z}}}[h(Z)] - D_{\mathrm{KL}}(Q_{\mathcal{Z}} \| P_{\mathcal{Z}}) \right) = \log \mathbb{E}_{Z \sim P_{\mathcal{Z}}}[\exp(h(Z))].$$

Since every distribution on $\mathcal{Z}$ is a pushforward of some distribution on $\mathcal{W}$ (i.e., for any $Q_{\mathcal{Z}} \ll P_{\mathcal{Z}}$, there exists $Q \in \mathcal{P}(\mathcal{W})$ ($\mathcal{P}(\mathcal{W})$ denotes the set of all probability measures on $\mathcal{W}$) such that $Q_{\mathcal{Z}} = \psi_\sharp Q$ and $P_{\mathcal{Z}} = \psi_\sharp P$ for some $P \in \mathcal{P}(\mathcal{W})$), and because $\mu \ll \nu$ implies $\psi_\sharp \mu \ll \psi_\sharp \nu$, we can rewrite the supremum over distributions in $\mathcal{W}$:

$$\sup_{Q \ll P} \left( \mathbb{E}_{Z \sim \psi_\sharp Q}[h(Z)] - D_{\mathrm{KL}}(\psi_\sharp Q \| \psi_\sharp P) \right) = \log \mathbb{E}_{Z \sim \psi_\sharp P}[\exp(h(Z))].$$

By the change of variables formula, the expectations and divergences can be pulled back to the weight space $\mathcal{W}$:

$$\sup_{Q \ll P} \left( \mathbb{E}_{X \sim Q}[h \circ \psi(X)] - D_{\mathrm{KL}}(\psi_\sharp Q \| \psi_\sharp P) \right) = \log \mathbb{E}_{X \sim P}[\exp(h \circ \psi(X))].$$

Thus, the variational formula in $\mathcal{Z}$ reduces to an expression entirely in terms of distributions and expectations over $\mathcal{W}$.

*Step 2: Applying to the relevant function.* For $\alpha > 0$, define

$$F : w \in \mathcal{W} \mapsto \alpha\big(L(f_w) - \hat{L}_S(f_w)\big).$$

Because $\psi$ factorizes $f : w \mapsto f_w$, it also factorizes $F$, so there exists $h$ such that $F = h \circ \psi$. Applying the lifted Donsker–Varadhan formula to $h$ gives:

$$\sup_{Q \ll P} \left( \mathbb{E}_{X \sim Q}[F(X)] - D_{\mathrm{KL}}(\psi_\sharp Q \| \psi_\sharp P) \right) = \log \mathbb{E}_{X \sim P}[\exp(F(X))].$$

*Step 3: Concentration inequalities on the prior.* At this point, we are in the same position as in the classical proofs of McAllester's PAC–Bayesian bounds (or any KL-based PAC–Bayesian bound). One can then follow the standard arguments (see, e.g., (Alquier, 2024, Theorem 2.1)), which mainly involve applying concentration inequalities (sub-Gaussianity of the loss and Chernoff bounds) to the prior term $\log \mathbb{E}_{X \sim P}[\exp(f(X))]$.

# B  BEYOND KL: CHANGE-OF-MEASURE TOOLS IN LIFTED SPACES

**Why this appendix.**  Section 3 establishes KL-based PAC-Bayes bounds *in lifted spaces*. The message here is broader: the same "lift-then-bound" template extends to other divergences that admit a change-of-measure principle. Divergences with this property used in known PAC-Bayes bounds include $f$-divergences (of which the KL divergence is a special case), Wasserstein distances. This matters because once a bound is valid in a lifted space, the **same computational challenges** reappear as in the KL case (Section 5): the complexity term becomes a divergence between $\psi_\sharp Q$ and $\psi_\sharp P$, which can be typically (i) tighter (e.g., by data processing), but also in general (ii) harder to compute. Hence, for each divergence, we face the same three-step agenda: *(validity - section 3)* prove a lifted change of measure, *(sharpness - section 4)* e.g. using DPI if applicable, and *(computation - section 5)* understand what is tractable in the chosen lifted space. Below, we discuss validity of PAC-Bayes bounds based on other complexity measure than the KL divergence.

**A generic lifted pattern.**  Let $D(\cdot\|\cdot)$ be a divergence endowed with a change-of-measure inequality that controls $\mathbb{E}_Q[L - \hat{L}_S]$, the generalization gap averaged over weights $w \sim Q$, in terms of $D(Q\|P)$ and an auxiliary term depending on $P$. If $\psi : \mathcal{W} \to \mathcal{Z}$ is measurable and is a lift, in the sense there is a function $g : \mathcal{Z} \to \mathcal{F}$ such that $f = g \circ \psi$ (factorization from Section 3), then the same argument in general applies with $Q, P$ replaced by $\psi_\sharp Q, \psi_\sharp P$, yielding a bound whose *complexity term* is $D(\psi_\sharp Q\|\psi_\sharp P)$. Moreover, whenever $D$ satisfies data processing,

$$D(\psi_\sharp Q\|\psi_\sharp P) \ \leq \ D(Q\|P),$$

it ensures the lifted bound is never looser at the level of the divergence term. The price to pay is computational: evaluating $D(\psi_\sharp Q\|\psi_\sharp P)$ can be more involved, exactly as we saw for KL.

## B.1  $f$-DIVERGENCE

For $f$-divergences $D_f(Q \| P) = \int_\Omega f(dQ/dP)dP$, a change of measure inequality exists. Specifically, for two probability distributions $Q, P$ such that $Q \ll P$, the following inequality holds (Nguyen et al., 2010; Picard-Weibel & Guedj, 2022; Polyanskiy & Wu, 2025):

$$D_f(Q \| P) = \sup_{g \text{ measurable}} (\mathbb{E}_Q[g] - \mathbb{E}_P[f^* \circ g])$$

where $f^*$ denotes the Fenchel conjugate of $f$. Similarly to Appendix A, this equality can be directly applied to the pushforward distributions $\psi_\sharp Q, \psi_\sharp P$:

$$D_f(\psi_\sharp Q \| \psi_\sharp P) = \sup_{g \text{ measurable}} (\mathbb{E}_{\psi_\sharp Q}[g] - \mathbb{E}_{\psi_\sharp P}[f^* \circ g])$$

Since $\psi$ is a lift, we can further rewrite the expectations in terms of the distributions $Q$ and $P$:

$$D_f(\psi_\sharp Q \| \psi_\sharp P) = \sup_{g \text{ measurable}} (\mathbb{E}_Q[g \circ \psi] - \mathbb{E}_P[f^* \circ g \circ \psi])$$

This form is particularly useful in the PAC-Bayes framework, as the expectation terms are expressed in terms of the weights, while the complexity term is evaluated in the lifted space. By the data-processing inequality (which holds for $f$-divergences (Polyanskiy & Wu, 2025, Theorem 7.4)), the complexity term in the lifted space is at least as sharp, leading to bounds that cannot degrade the usual ones.

**Takeaway.**  All PAC-Bayes bounds derived from $f$-divergences admit a lifted counterpart with a divergence term that can be only smaller. As in the KL case, the remaining question is *computability* of $D_f(\psi_\sharp Q\|\psi_\sharp P)$ for the chosen lift $\psi$.

## B.2  WASSERSTEIN DISTANCES

PAC-Bayes bounds based on Wasserstein distances rely on the change-of-measure inequality provided by Kantorovich–Rubinstein duality (Villani, 2009, Theorem 5.9). For the 1-Wasserstein distance (with $P, Q$ in the Wasserstein space of order 1 $\mathcal{P}_1(\mathcal{W}) := \{\mu \text{ proba on } \mathcal{W} \text{ s.t. } \int_{\mathcal{W}} \|w\|_1 d\mu(w) < \infty\}$),

$$\kappa \, W_1(Q, P) \ = \ \sup_{\|h\|_{\text{Lip}} \leq \kappa} \left( \mathbb{E}_Q[h] - \mathbb{E}_P[h] \right).$$

This immediately implies

$$\mathbb{E}_Q[L - \hat{L}_S] - \mathbb{E}_P[L - \hat{L}_S] \leq \kappa_{\mathcal{W}} W_1(Q, P)$$

as soon as the map $w \mapsto (L - \hat{L}_S)(w)$ is $\kappa_{\mathcal{W}}$-Lipschitz in weight space. However, in practice, known upper bounds on the Lipschitz constant in weight space are usually very loose. The most classical one scales as the product of spectral norms of the layers, which can grow exponentially with depth and make the resulting bound vacuous.

To obtain a similar bound with the Wasserstein distance between the *lifted* distributions $\psi_\sharp Q$ and $\psi_\sharp P$, one must[5] therefore show that the generalization gap is Lipschitz in the lifted representation. This question is well-posed: the loss depends on the weights $w$ only through the function $f_w$ implemented by the network, and since $f_w$ can be written as $g \circ \psi(w)$ for some suitable $g$ (e.g. in path-based parametrizations), it follows that there exists a (possibly ugly) function $h$ such that

$$(L - \hat{L}_S)(w) = h(\psi(w)).$$

In other words, the generalization gap depends on $w$ only through its lifted coordinates $z = \psi(w)$. If the map $z \mapsto h(z)$ is itself Lipschitz, then Kantorovich–Rubinstein duality directly yields a Wasserstein-based PAC-Bayes bound in lifted space.

Here lies a key difference with KL (and more generally $f$-divergences): for KL, the bound in the lifted space follows *automatically* from the factorization of the generalization gap through $\psi$; for Wasserstein, the lift must in addition preserve Lipschitzness.

The path+sign lift studied in Section 5 provides precisely such a property: the network output is known to be Lipschitz in the path-lifting representation on each closed orthant of $\mathcal{W}$ (Gonon et al., 2025, Thereom 4.1). Since standard losses are themselves Lipschitz in the network outputs, this implies that the loss gap is Lipschitz in the lifted coordinates, at least when restricted to a single orthant. This suggests the following template.

**Informal Statement 5** (lifted $W_1$ control under orthant-wise Lipschitzness)**.** *Assume there exists a lift $\psi : \mathcal{W} \to \mathcal{Z}$ and a constant $\kappa_{\mathcal{Z}} > 0$ such that $z \mapsto (L - \hat{L}_S)(g(z))$ is $\kappa_{\mathcal{Z}}$-Lipschitz on each orthant of $\mathcal{W}$. If $Q$ and $P$ are both supported on the same orthant (e.g., by conditioning on signs), then*

$$\mathbb{E}_Q[L - \hat{L}_S] - \mathbb{E}_P[L - \hat{L}_S] \leq \kappa_{\mathcal{Z}} W_1(\psi_\sharp Q, \psi_\sharp P).$$

In summary, the Wasserstein case illustrates well the three-step agenda of lifting divergences to intermediary spaces between the function space and weight space.

*(i) Validity.* Thanks to the factorization $(L - \hat{L}_S)(w) = h(\psi(w))$, it makes sense to ask whether the generalization gap is Lipschitz in the lifted space. For path-based lifts enriched with signs, this is indeed the case on each closed orthant [6], that is, on each region of the weight space where the sign of every coordinate is fixed (including the boundaries where some coordinates may be zero). So the basic validity of a lifted Wasserstein bound is established.

*(ii) Improvement.* Unlike KL (where improvement is guaranteed by the data processing inequality), here both sides of the inequality change: the divergence $W_1(Q, P)$ becomes $W_1(\psi_\sharp Q, \psi_\sharp P)$, and the Lipschitz constant $\kappa_{\mathcal{W}}$ becomes $\kappa_{\mathcal{Z}}$. Known bounds on $\kappa_{\mathcal{Z}}$ [7] for the path-lifting are still large, but they are provably less pessimistic (sometimes dramatically so) than the naive weight-space bound on $\kappa_{\mathcal{W}}$ given by the product of spectral norms Gonon (2024). This indicates that lifting can mitigate part of the curse of depth of usual Lipschitz constants, and could help to improve Wasserstein-based

---

[5]And one should also check that the lifted distributions $\psi_\sharp Q$ and $\psi_\sharp P$ are in the Wasserstein space of order 1 denoted by $\mathcal{P}_1(\mathcal{W})$. This is true for the lift $\psi = (\Phi, \text{sign})$ based on the path-lifting $\Phi$, as in Section 5, for every $P, Q \in \mathcal{P}_1(\mathcal{W})$ that factorizes along the coordinates $w_i$ (i.e., such that the coordinates are independents). Indeed, consider $\mu = \otimes_{i=1}^{\dim(\mathcal{W})} \mu_i$ a probability distribution on $\mathcal{W}$, then using $|\text{sign}| \leq 1$ and the definition of $\Phi$, we get $\int_{\psi(\mathcal{W})} \|\psi(w)\|_1 \mathrm{d}\psi \sharp \mu(w) \leq \int_{\psi(\mathcal{W})} \|\Phi(w)\|_1 \mathrm{d}\psi \sharp \mu(w) + 1 = \sum_{\text{paths } p} \prod_{i \in p} \int_{\mathcal{W}_i} |w_i| \mathrm{d}\mu(w_i) + 1 < \infty$.

[6]This follows from the Lipschitz property of the realization function $f$. with respect to the lift $\psi$, which carries over to the generalization error (see Viallard et al. (2023); Haddouche & Guedj (2023) for two approaches).

[7]Which are derived from the bounds on the Lipschitz constant of the realization function $f$. with respect to the lift $\psi$.

bounds. However, the divergence term itself can also increase under lifting: for instance, in the case of Dirac measures, one may encounter situations where

$$\|w - w'\| \leq \|\psi(w) - \psi(w')\|,$$

so that the Wasserstein distance grows after lifting. For instance, consider two weight vectors: $w = (3, 3)$ and $w' = (0, 0)$, representing the weights of a one-hidden-neuron neural network. We have $\|w - w'\|_1 = 6$, but $\|\psi(w) - \psi(w')\|_1 = \|\Phi(w) - \Phi(w')\|_1 + \|\operatorname{sign}(w) - \operatorname{sign}(w')\|_1 = |9 - 0| + |1 - 0| + |1 - 0| = 11$. This stands in stark contrast to the KL divergence case, where such an increase is precluded by the data processing inequality.

*(iii) Practicality.* Two difficulties remain before such bounds become usable in practice: extending orthant-wise arguments to handle sign changes, and computing high-dimensional Wasserstein distances between lifted distributions. These mirror the challenges already encountered for KL in Section 5: lifting sharpens the complexity term in principle, but turning this into tractable, non-vacuous guarantees requires further structural insights.

## C  PROOF OF THEOREM 4

We use as notations $\operatorname{ant}(v), \operatorname{suc}(v)$ for antecedents/successors of a neuron $v$ in the graph (in/out neighbors), $D_{\mathrm{KL}}(\cdot\|\cdot)$ for Kullback–Leibler and $D(\cdot\|\cdot)$ for a general divergence satisfying the data-processing inequality $D(F_\sharp Q \| F_\sharp P) \leq D(Q\|P)$ for any $Q, P$ and any pushforward $F$, and $\diamond^\lambda$ for the neuron-wise rescaling action defined in (5).

### C.1  PROBLEM REDUCTION (TWO-SIDED TO ONE-SIDED)

Denoting $\Lambda$ the diagonal matrix such that $\diamond^\lambda(w) = \Lambda w$ for every $w$, and similarly $\Lambda'$ such that $\diamond^{\lambda'}(w) = \Lambda' w$, we have $\diamond^\lambda_\sharp Q = \Lambda_\sharp Q$ and $\diamond^{\lambda'}_\sharp P = \Lambda'_\sharp P$. From the well-known group structure of rescaling invariances both $\Lambda$ and $\Lambda'$ are invertible and there exists $\hat\lambda$ such that $\hat\Lambda := \Lambda'^{-1}\Lambda$ is a diagonal matrix such that $\diamond^{\hat\lambda}(w) = \hat\Lambda w$. Since the data processing inequality (DPI) implies the *equality* $D(Q\|P) = D(F_\sharp Q \| F_\sharp P)$ for any distributions whenever $F$ is an invertible function (DPI applied to $F$ and to $F^{-1}$ gives both $\leq$ directions), we obtain that $D(\diamond^\lambda_\sharp Q \| \diamond^{\lambda'}_\sharp P) = D(\Lambda_\sharp Q \| \Lambda'_\sharp P) = D((\Lambda'^{-1}\Lambda)_\sharp Q \| P) = D(\diamond^{\hat\lambda}_\sharp Q \| P)$, hence the result with $\bar{J}(\lambda) := D(\diamond^\lambda_\sharp Q \| P)$. A similar reasoning yields the result with $J(\lambda) = D(Q \| \diamond^\lambda_\sharp P)$.

### C.2  EXISTENCE AND UNIQUENESS OF THE GLOBAL MINIMIZER

We now focus on the KL divergence and a centered Gaussian prior $P = \mathcal{N}(0, \sigma'^2 \mathbf{I})$ (the proof easily extends to arbitrary diagonal covariance for $P$), assuming also that the posterior $Q$ admits a density with respect to the Lebesgue measure, and has finite second moments.

With the rescaling vector $\lambda \in \mathbb{R}^{|H|}_{>0}$ and the corresponding diagonal matrix $\Lambda$ as above, observe that $\diamond^\lambda_\sharp P = \Lambda_\sharp P = \mathcal{N}(0, \sigma'^2 \Lambda^2)$ so that for any vector $w$

$$f_\lambda(w) := -\log \diamond^\lambda_\sharp P(w) = \frac{\|\Lambda^{-1}w\|_2^2}{2\sigma'^2} + \log \det \Lambda + c$$

for some constant $c$ that will be irrelevant when optimizing $J(\lambda)$. It follows that

$$J(\lambda) = D_{\mathrm{KL}}(Q\| \diamond^\lambda_\sharp P) = \mathbb{E}_{w \sim Q}[-\log \diamond^\lambda_\sharp P(w)] - \mathbb{E}_{w \sim Q}[-\log Q(w)]$$

$$= \frac{1}{2\sigma'^2}\mathbb{E}_{w \sim Q}\|\Lambda^{-1}w\|_2^2 + \log\det\Lambda + c'$$

$$= \frac{1}{2\sigma'^2}\underbrace{\sum_{e \in E}\left(\Lambda_{ee}^{-2}\sigma_e^2 + 2\sigma'^2 \log \Lambda_{ee}\right)}_{=:\hat{J}(\lambda)} + c'$$

where the sum is over edges of the graph $G = (V, E)$ and $\sigma_e^2 := \mathbb{E}_{w \sim Q} w_e^2$ is the variance of the weight on the edge indexed by $e$ (note that we have used above that $Q$ has finite second order moments and is absolute continuous w.r.t. $P$).

As detailed below, considering $z = \log \lambda \in \mathbb{R}^H$ we can express $\Lambda$ as $\Lambda = \mathrm{diag}(\exp(Bz))$ (see details below) where logarithms and exponentials are entrywise and $B$ is some matrix with linearly independent columns associated to the DAG structure of the considered network. Denoting $b_e$ the $e$-th row of $B$ we thus have $\Lambda_{ee} = \exp(\langle b_e, z \rangle)$, and optimizing $J(\lambda)$ is equivalent to optimizing $\hat{J}(\lambda)$ or equivalently as a function of $z$ (which we still denote by $\hat{J}$ by slight abuse of notations):

$$\hat{J}(z) := \sum_e \left( \sigma_e^2 e^{-2\langle b_e, z \rangle} + 2\sigma'^2 \langle b_e, z \rangle \right). \tag{13}$$

As a sum of strictly convex continuous functions, $\hat{J}(z)$ is continuous and strictly convex, and since the columns of $B$ are linearly independent there is a constant such that $\max_e |\langle b_e, z \rangle| = \|Bz\|_\infty \geq c\|z\|$, hence $\hat{J}(z)$ is also coercive. This shows the existence and uniqueness of a global minimizer.

**Expressing $\Lambda$ as a function of $z = \log \lambda$.**  The key identity is that if $e = u \to v$ is an edge (from neuron $u$ to neuron $v$) then

$$(\Lambda w)_e := (\diamond^\lambda(w))_e = \frac{\lambda_v}{\lambda_u} w_e = \exp(z_v - z_u) w_e$$

hence $\Lambda_{ee} = \exp(z_v - z_u)$. This yields the result where $B$ is the matrix with entries

$$B_{eh} := \begin{cases} 1, & \text{if } e = u \to h \text{ for some } u \in V \\ -1, & \text{if } e = h \to v \text{ for some } v \in V \\ 0 & \text{otherwise.} \end{cases}$$

It can be checked that $B$ has linearly independent columns.

## C.3 CONVERGENCE OF THE BCD SCHEME

By (13) and explicit expression of $B$, we expand $\hat{J}(z)$ as a sum of edgewise univariate functions

$$\hat{J}(z) = \sum_{v \notin V_{\mathrm{in}}} \sum_{u \in \mathrm{ant}(v)} \left( \sigma_{u \to v}^2 e^{-2(z_v - z_u)} + 2\sigma'^2 (z_v - z_u) \right).$$

By the global coercivity of $\hat{J}$, its level sets are compact, and by its strict convexity, each one-dimensional block section $t \mapsto \hat{J}(z_0 + tz_1)$ has a unique minimizer with a closed-form expression that we will explicit below. By Tseng's essentially cyclic BCD theorem (Tseng, 2001, Thm. 4.1) (see also Stock et al. (2019) for a related use), we conclude that the iterates converge, and combine with uniqueness to get convergence to $z^\star = \log \lambda^\star$.

We now seek one-dimensional minimizers on some coordinate indexed by $v_0 \in H$. Since

$$\hat{J}(\lambda) = \sum_{v \notin V_{\mathrm{in}}} \sum_{u \in \mathrm{ant}(v)} \left( \sigma_{u \to v}^2 (\lambda_u / \lambda_v)^2 + 2\sigma'^2 \log(\lambda_v / \lambda_u) \right),$$

when fixing the values $\lambda_u$, $u \neq v_0$ and optimizing over the remaining variable $\lambda_{v_0}$, the function to be optimized writes (up to a constant independent of $\lambda_{v_0}$) as

$$A\lambda_{v_0}^2 + C\lambda_{v_0}^{-2} + 2B \log \lambda_{v_0} = F(\lambda_{v_0}^2) \text{ with } F(X) := AX + C/X + B \log X$$

where

$$A = A_{v_0}(\lambda) := \sum_{v \in \mathrm{suc}(v_0)} \frac{\sigma_{v_0 \to v}^2}{\lambda_v^2}, \tag{14}$$

$$C = C_{v_0}(\lambda) := \sum_{u \in \mathrm{ant}(v_0)} \sigma_{u \to v_0}^2 \lambda_u^2, \tag{15}$$

$$B = B_{v_0} := \sigma'^2 \left( \sharp \mathrm{ant}(v_0) - \sharp \mathrm{suc}(v_0) \right). \tag{16}$$

Minimizing over $\lambda_{v_0} \in \mathbb{R}_{>0}$ amounts to minimize $F(X)$ over $X > 0$, which reduces to finding a positive root of its derivative, which is a positive root of the quadratic equation $AX^2 + BX - C = 0$. This yields

$$X_{v_0}^\star(\lambda) := \frac{-B + \sqrt{B^2 + 4AC}}{2A} \tag{17}$$

$$\lambda_{v_0}^\star(\lambda) := \sqrt{X_{v_0}^\star(\lambda)} \tag{18}$$

**Remark (orders and parallel schedules).** The proof above uses single-coordinate updates in any essentially cyclic order (e.g., a topological order repeated). For LFCNs, neurons in the same layer are independent given their neighbors, which permits parallel layerwise updates; moreover, the odd–even (red–black) schedule is an essentially cyclic scheme and thus inherits the same convergence guarantee.

**Treating biases (optional).** If biases are used, append a constant-1 input neuron and interpret the bias of a neuron $v$ as the weight of the edge going from the constant-1 input neuron to $v$. In particular, this augments the set of predecessors of $v$ by one in Equations (14) to (16).

**Case of square LFCN without bias** When $Q = \mathcal{N}(\mu, \sigma^2 \mathbf{I})$ and the network is an LFCN without biases we have $B = 0$ (each hidden neuron has as many incoming weights than ougoing weights). This yields the simple expression in (11).

## D EXPERIMENTAL DETAILS

All experiments were conducted on a MacBook Pro (M4, 2025) using PyTorch 2.7.0.

**MNIST** Models were trained with SGD (learning rate $0.1$, no weight decay, batch size $256$), using a Gaussian prior ($\mu = 0$, $\sigma = 1$) and a posterior defined by the trained weights as mean and a fixed standard deviation $\sigma = 0.03$, selected via preliminary sweeps on the values of $\sigma$ using the sweep agent of the Python library wandb. PAC-Bayes bounds were computed using McAllester's bound with confidence parameter $\delta = 0.05$. The total compute time for the sweep was 33 minutes (10 runs, approximately 3 minutes per run). Trained models and raw results will be released in a non-anonymous repository upon acceptance.

**CIFAR-10** For CNN experiments, we used the architecture introduced in Gitman & Ginsburg (2017), which consists of 9 convolutional layers and 3 pooling layers, without batch normalization. The model was trained following the protocol described in the original paper: SGD with a learning rate linearly decayed from $0.01$ to $10^{-5}$, a weight decay of $0.002$, a batch size of $128$, and for a total of 50 epochs. We employed a zero-mean Gaussian prior ($\mu = 0$) and centered the posterior on the trained weights. The prior standard deviation $\sigma_{\text{prior}}$ was sampled uniformly from the interval $[0.01, 1]$, while the posterior standard deviation $\sigma_{\text{posterior}}$ was sampled uniformly from $[0.0001, 0.05]$ through a random sweep. The total training time for this model was 34 minutes, and the sweeper required 4 hours to compute the different PAC-Bayes bounds.

## E USE OF LLMS

We made limited use of large language models during the preparation of this manuscript. Their role was strictly restricted to grammar correction, improving clarity and conciseness and emphasizing text (e.g., bolding). They were not used for generating technical content, suggesting new concepts, or contributing to proofs or results. All ideas, proofs, experiments, and findings are entirely our own. Every rephrased passage was carefully reviewed and validated by the authors to ensure correctness and faithfulness to our original intent. No unverified or plagiarized content was introduced.

