# OpenReview forum: "Non-Vacuous Generalization Bounds: Can Rescaling Invariances Help?"
_ICLR.cc/2026/Conference — ICLR 2026 Conference Withdrawn Submission_

### Official Review · Reviewer_Ypnt · 2025-11-01

**Soundness:** 3
**Presentation:** 2
**Contribution:** 2
**Rating:** 4
**Confidence:** 3

**Summary:**

The elephant in the room of PAC-Bayes bounds is  \textbf{rescaling invariances}. Different weight configurations can represent the exact same function but yield vastly different PAC-Bayes complexities.

This paper proposes addressing this issue by two routes: 1) Optimise  PAC-Bayes bounds directly in weight space by minimising over deterministic or stochastic rescalings of the weights, 2) Define and analyse PAC-Bayes bounds in an invariant, lifted space, where parameters that represent the same function are identified.

In this way they improve the performance of PAC-Bayes bounds, showing some examples for MNIST and CIFAR10.

**Strengths:**

They identify the key problem with PAC-Bayes bounds: the need to incorporate natural invariances.

They provide a method to improve PAC-Bayes bounds that seems to work, occasionally even leading to non-vacuous bounds, at least for the settings they study.

**Weaknesses:**

The methods they use are rather computationally involved.  That is not fatal, but it would be good if they were easier to implement.

However, I have a more fundamental worry about these kinds of approaches. The aim of generalisation bounds in deep learning is not really guarantees; it's much easier to use other methods for that.  Instead, we should use them to gain insight into what drives generalisation.  The difficulty with very complex approaches of the kind shown in this paper is that we don't learn much about what makes neural networks generalise well.

Secondly, the authors miss some key papers that they need to engage with; it's unclear to me what exactly is truly novel compared to
say Scale-invariant Bayesian Neural Networks with Connectivity Tangent Kernel https://arxiv.org/abs/2209.15208  which also provides tighter bounds with scale-invariant methods, but also provides some new insight into a better proxy for generalisation.

Also, there are already good non-vacuous PAC-Bayes bounds such as the marginal-likelihood bound from https://arxiv.org/abs/2012.04115 that naturally takes into account scale-invariances.  The authors don't engage with this older work.

Finally, given that one wants a generalisation measure such as a PAC-Bayes bound to explain something, such bounds should be tested in a wider range of settings, e.g. for different training set sizes, e.g. learning curves -- can this bound reproduce a scaling exponent as done in https://arxiv.org/abs/2012.04115  ?

**Questions:**

How do your bounds relate to the non-vacuous bounds  in https://arxiv.org/abs/2209.15208 -  I suspect your definition of "lifting" is a generalization of the function space view PACBayes, which this paper already considered, providing a similar upper bound (function space KL is strictly <= parameter space KL).

Can this bound reproduce a scaling exponent as done in https://arxiv.org/abs/2012.04115  ?


How do your bounds relate to  https://arxiv.org/abs/2209.15208  -- what is the big step forward in your work compared to this paper?  Can you similarly extract some practical insight, as they do for uncertainty calibration ?

---

### Official Review · Reviewer_3Aae · 2025-11-02

**Soundness:** 3
**Presentation:** 3
**Contribution:** 1
**Rating:** 2
**Confidence:** 4

**Summary:**

The paper studies PAC Bayes type generalization bounds, particularly in the context of deep learning, and the behavior of these bounds under natural rescalings of the network weights. To recall, PAC Bayes generalization bounds aim to express generalization in terms of information theoretic quantities such the KL divergence between the posterior distribution and the prior (both on the set of weights) and has gained recent (and not so recent) interest in deep learning theory as a potential avenue for reasoning about generalization in deep neural networks. The starting point of the present paper is that the weights of the neural netwrks don't uniquely specify the function that they express and there are inherent rescaling invariants in these models. The aim question that the paper aims to address is whether the invariances can be used to provide improved generalization bounds in the PAC bayes setting. The paper provides bounds that take into account scaling invariances and get improved PAC Bayes bounds under the appropriate scaling invariances.

**Strengths:**

The broad question considered by the paper is an interesting one. I believe that incorporating invariances in generalization is an important aspect in eventually understanding generalization. With this regard, the questions considered by this paper is natural.

**Weaknesses:**

That said, I don't find the theoretical results presented in the paper particularly interesting or insightful. In particular, I suspect the overall approach is "folklore" in the community. I say this not to say that the results are not "difficult" to prove but just because the techniques presented are fairly common knowledge in the community. That said, had this perspective been especially useful in provide heretoforth unseen generalization bound that are "nontrivial" in either theory or practice, my criticism would have been moot. But the paper does not seem to present any evidence in that direction.

**Questions:**

-> The approach presented (at least the intial abstract theory) seems to be general for all invariances. any reason to restrict to scaling other the optimization aspect later in the paper?
-> Do you think that there are simple experiments that one could run (maybe in toy settings) that would indicate that this perspective gives improved bounds? Unfortunately, most simple toy settings, I imagine either this would be "tautotogical true" (i.e. settings are scaling artificially increases complexity) or even the invariant version of PAC Bayes does not explain generalization sufficiently? If indeed there is an interesting experiment indicating this I would be interested to see this included.

---

### Official Review · Reviewer_5t3P · 2025-11-03

**Soundness:** 2
**Presentation:** 3
**Contribution:** 1
**Rating:** 2
**Confidence:** 3

**Summary:**

The authors revisit PAC-Bayes generalization theory through the lens of rescaling invariances in ReLU networks. Standard PAC-Bayes bounds are expressed in weight space, which ignores symmetries such as neuron-wise rescaling that leave the function unchanged but can arbitrarily increase the KL divergences. The paper introduces a lifted representation that collapses these symmetries and provides invariant, potentially tighter generalization bounds. It develops two complementary approaches which are the optimization over deterministic or stochastic rescalings in weight space, as well as a formal lift to an invariant latent space. The authors establish a chain of inequalities showing that lifted and rescaling-optimized KLs are never worse and often strictly better than standard ones, and they propose a tractable deterministic rescaling algorithm. Empirically, the method halves PAC-Bayes bound values for MLPs on MNIST and CNNs on CIFAR-10, turning some vacuous bounds for MNIST into non-vacuous. However, the CIFAR-10 bounds remain vacuous, despite the proposed interventions.

**Strengths:**

- The authors clearly identify a key gap in existing PAC-Bayes bounds for invariant settings, highlighting the misalignment between parameter-space divergences and function-space equivalences, and offers two ways to overcome this issue.
- The lifted-space PAC-Bayes derivation is rigorous, cleanly extending the Donsker–Varadhan argument.
- The deterministic rescaling proxy is a useful practical step.

**Weaknesses:**

- While I appreciate the references existing throughout the paper, the paper needs a prior related work section discussing how this work relates to existing literature on exploiting redundancies and invariances to make bounds tighter, e.g. pruning being one way to break the redundancy and make the bounds tighter as in https://arxiv.org/abs/1804.05862 [1].
- The empirical scope is quite limited as the experiments are done at the small scale (MNIST, CIFAR-10).
- The biggest weakness of this work is that it does not establish the practical usefulness of the proposed approach. For MNIST, the bounds obtained appear worse than state-of-the-art PAC-Bayes bounds given a fixed dataset (0.11 for data-independent and 0.14 for data-dependent priors -> see [2] https://arxiv.org/abs/2312.17173). For CIFAR-10, the bounds remain vacuous. So while the theory is rigorous and informative, it does not yet lead in practice to tight, non-vacuous PAC-Bayes bounds for datasets beyond MNIST or for large-scale models, as stated to be the ambition of the paper.

Very minor:
- The abstract could better distinguish the lifted vs. deterministic rescaling contributions.
___
References:

[1] Zhou, W., Veitch, V., Austern, M., Adams, R.P. and Orbanz, P., 2018. Non-vacuous generalization bounds at the imagenet scale: a PAC-bayesian compression approach. arXiv preprint arXiv:1804.05862.

[2] Lotfi, S., Finzi, M., Kuang, Y., Rudner, T.G., Goldblum, M. and Wilson, A.G., 2023. Non-vacuous generalization bounds for large language models. arXiv preprint arXiv:2312.17173.

**Questions:**

- What is the exact value of the MNIST bound? How do they compare to other approaches that try to obtain tight bounds through compression for instance by projecting to a linear subspace?
- Can you obtain non-vacuous bounds for CIFAR-10 by trying different architectures, hyperparameters ..etc? That would make for a better empirical demonstration?
- For larger models, does the rescaling optimization remain stable numerically, or do the λ updates diverge?

See weaknesses for other remarks.

---

### Official Review · Reviewer_WE9b · 2025-11-17

**Soundness:** 3
**Presentation:** 4
**Contribution:** 2
**Rating:** 4
**Confidence:** 3

**Summary:**

The authors study the effect of invariances on generalization bounds for deep learning. In particular, the paper focuses on PAC-Bayes bounds and analyzes the effect of weight rescaling. The authors derive bounds that account for scaling invariances using two distinct approaches. In doing so, they improve the theoretical performance of PAC-Bayes bounds and provide examples on MNIST demonstrating that their method can transform previously vacuous bounds into non-vacuous ones.

**Strengths:**

- The question considered in this paper is important, namely incorporating invariances into generalization bounds in deep learning.
- The motivations behind studying the effect of rescaling of weights in generalization bounds in deep learning are well-justified in the paper.
- The paper provides original contributions.
- I have verified most of the proofs, and they appear to be correct.
- The paper includes a few empirical results demonstrating the advantage of the proposed approach.
- Overall, the paper is well-written and easy to follow.

**Weaknesses:**

- The discussion of related work should be strengthened. There is a substantial body of literature on generalization bounds for neural networks that the paper does not cover. Btw, I do not agree with the claim that PAC-Bayes bounds stand out among existing approaches.
- My main concern is that the proofs appear relatively straightforward. I do not see a substantial technical contribution.

**Questions:**

- Is there a novel thereotcial aspect I might have overlooked, or a specific obstacle in the proofs that required new ideas to overcome?

**Details Of Ethics Concerns:**

-

---

### Author Response · Authors · 2025-12-02

We thank all the reviewers for their time and feedback. Your comments helped us identify several aspects of positioning and presentation that we believe deserve a careful treatment. To best reflect the scope and contribution of our work, we have decided to take the necessary time to reorganize the paper and strengthen its framing and empirical narrative. For this reason, we are withdrawing the submission and look forward to resubmitting an improved version in the future.

---

### Note · Authors · 2025-12-02

I have read and agree with the venue's withdrawal policy on behalf of myself and my co-authors.